# A High-Detection-Efficiency Optoelectronic Device for Trace Cadmium Detection

**DOI:** 10.3390/s22155630

**Published:** 2022-07-28

**Authors:** Huangling Gu, Long Wang

**Affiliations:** 1School of Metallurgy and Environment, Central South University, Changsha 410083, China; guhuangling@csu.edu.cn; 2School of City and Environment, Hunan University of Technology, Zhuzhou 412007, China

**Keywords:** cadmium, spectral measurement, photodetector

## Abstract

Cadmium (Cd) pollution in soil is a serious threat to food security and human health, while, currently, the most widely used detection methods cannot accurately reflect the content of heavy metals in soil. Soil heavy metal detection combined with microelectronic sensors has become an important means of environmental heavy metal pollution prevention and control. X-ray Fluorescence spectrometry (XRF) can capture the excitation spectrum of metal elements, which is often used to detect Cd (II). However, due to the lack of high-performance optoelectronic devices, the analysis accuracy of the system cannot meet the requirements. Therefore, this study proposes a high-detection-efficiency photodiode (HDEPD) which can effectively improve the detection accuracy of the analyzer. The HDEPD is manufactured based on a 0.18 μm standard complementary metal-oxide-semiconductor (CMOS) process. The volt-ampere curve, spectral response and noise characteristics of the device are obtained by constructing a test circuit combined with a spectral detection system. The test results show that the threshold voltage of HDEPD is 12.15 V. When the excess bias voltage increases from 1 V to 3 V, the spectral response peak of the device appears at 500 nm, and the photon detection probability (PDP) increases from 41.7% to 52.8%. The dark count rate (DCR) is 31.9 Hz/μm^2^ at a 3 V excess bias voltage. Since the excitation spectrum peak of Cd (II) is between 500 nm and 600 nm, the wavelength response range of HDEPD fully meets the detection requirements of Cd (II).

## 1. Introduction

Cadmium is a toxic substance. The accumulation of heavy metal Cd in soil can pollute crops and endanger human health through the food chain. Additionally, long-term exposure may cause acute and chronic poisoning of the human body, even leading to teratogenic, carcinogenic and mutagenic risks [1]. Therefore, the accurate and efficient determination of soil Cd content, effective comprehensive prevention, and control of Cd-contaminated soil are the focus of attention worldwide [2]. Cd is usually present in amounts less than 1 mg/kg because of its low content in soil [1,2]. At present, the common methods used for detecting cadmium include atomic absorption spectrometry (the limit of detection of Cd (II) in soil is generally 0.0003~0.0089 mg/kg, with the recovery rate of 89.4~90.9% and the relative standard deviation of 0.8%) [3,4], spectrophotometry (the limit of detection of Cd (II) in soil is generally 0.01~0.02 mg/kg, with the recovery rate of 83.2~91.9% and the relative standard deviation of 3.8%) [5,6], plasma mass spectrometry (the limit of detection of Cd (II) in soil is generally 0.003~0.008 mg/kg, with the recovery rate of 97.3% and the relative standard deviation of 3.81%) [7,8], etc. Although these methods are sensitive and accurate, they have the disadvantage of requiring complex pretreatments, long analysis times, and large instrument volumes; damaging samples; and having high prices and high maintenance requirements [9]. With the progress of science and technology, new detection methods include electrochemical analyses and biological analyses, which have the characteristics of simple operation, low cost, rapid detection, and being suitable for on-site detection. However, there are some problems, such as the cumbersome sample pretreatment needed, low sensitivity, and fewer field applications. Therefore, electrochemical analyses and biological analyses have not been popularized yet [10,11]. The current mainstream is optical analysis. Compared with traditional optical analysis methods, detection methods based on spectra such as X-ray Fluorescence spectrometry (XRF, the limit of detection of Cd (II) is generally 2~5 mg/kg) [12,13,14] and visible near-infrared spectrum (Vis-NIR, characteristic bands: 400~600 nm, 979 nm, 2206 nm; relative error: 30~40%) [15,16] are characterized by rapid, simple, portable, and non-destructive testing [17,18]. However, due to its high limit of detection and certain errors, it is difficult to obtain obvious spectral lines when using spectral technology to analyze the Cd content in soil. Improving the accuracy of Cd spectral detection is of great significance in the application of soil heavy metal detection. Nowadays, the spectral non-destructive testing method based on microelectronic process integration has the advantages of rapid detection, simplicity, high resolution, and multiple bands [19,20,21] and has become an important method for the real-time and efficient detection of environmental pollution. For example, the detection of Pb (II) and Hg (II) in water can use three nickel hexacyanoferrate-film-modified Pt/n-n+-si electrodes to fulfil high-precision detection requirements [22]; the limit of detection of paclobutrazol in soil can be 0.075 mg/kg using the sensor based on Ag-based composite surface enhanced Raman scattering (SERS) substrate [23,24]; and the application of silicon drift detector (SDD) in XRF can cause the limit of detection of the Cd (II) content in soil reach 0.2 mg/kg and the qualified rate of accuracy is greater than 90%, which can meet the requirements of soil pollution detection [25].

In response to the above application requirements, many scholars have carried out research on high-performance photodetectors. Using single-photon detection technology can greatly improve the detection efficiency of optoelectronic devices, such as photomultiplier tubes (PMTs) [26,27], charge-coupled devices (CCDs) [28], and avalanche photodiodes (APDs) [29,30]. However, PMT and CCD are not compatible with the standard integrated circuit design process, which limits the application of these devices. APD has a mature theoretical basis and has become the preferred scheme for system integration. In addition, according to the working requirements of the photosensitive surface of APD, designing an appropriate guard ring can effectively enable the device to work in Geiger mode—that is, a single-photon avalanche diode (SPAD). The structural design of SPAD is mainly inspired by the research of Haitz et al. [31]. This scholar constructed a PN junction on a silicon substrate and skillfully used an N-type low-concentration region to form a guard ring to prevent avalanche breakdown at the edge of the space charge region, which provided a reference for the design of subsequent SPAD. Kim et al. formed a SPAD device with N+ and P-Well and verified it based on a high-voltage CMOS process. The test results showed that the avalanche breakdown voltage of the device was 10.7 V, and the quantum efficiency could reach 30% at a wavelength of 650 nm [32]. In order to achieve a low-noise SPAD structure, Yang et al. designed a device with an active region diameter of 8 μm using a 0.18 μm standard CMOS. At room temperature, the threshold voltage of SPAD is 14.2 V, and that of the dark count rate (DCR) is 260 Hz under a 2 V excess bias voltage. This structure can provide enlightenment for device design for a low dark count [33]. Shallow trench isolation (STI) performed by the standard process can significantly improve the post-pulse effect of SPAD because the energy level defects in STI will trap noisy electrons and cause DCR multiplication. Therefore, optimizing the guard ring of SPAD has also become an important research direction in device design [34,35,36]. Gersbach et al. studied the defect principle of STI and optimized the surface traps of the isolation layer by passivation implantation, showing that the DCR of SPAD under a 2 V excess bias voltage was only 3.79 Hz/μm^2^ [37]. Webster et al. realized a near-infrared responsive SPAD based on a 0.13 μm standard microelectronic process. By optimizing the structure and increasing the reverse excess bias voltage, the device could achieve a more than 40% photon detection probability (PDP) in the wavelength range of 410 nm~760 nm [38]. The above research laid a solid structural foundation for the design of SPAD [39,40,41,42]. However, for the spectral selectivity requirements of Cd (II) detection, there is a lack of device structures with a high detection efficiency in the wavelength range of 500~600 nm. Therefore, a photodiode with a high detection efficiency is proposed in this work. The structure is fully compatible with the standard microelectronic processes, and the system integration can be realized without any mask correction. By designing a suitable depletion layer, HDEPD can achieve a high photon detection probability at the wavelength of 500 nm, meaning that it can effectively meet the high-precision detection requirements of Cd (II) according to the US NIST atomic spectroscopy database [15].

## 2. Structure and Operating Principle of the High-Detection-Efficiency Photodiode

The three-dimensional structure and cross-section of the high-detection-efficiency photodiode device are shown in Figure 1a,b, respectively. The N+ ring injection regions on the left and right sides of the HDEPD are the cathode of the device, and the P+ ring injection region and the polysilicon ring gate in the middle are the anode of the device. When the potential difference between the cathode and anode of the device is greater than the avalanche breakdown threshold of P+/NW, the HDEPD operates in Geiger mode. At this time, the photon signal with an energy greater than the band gap of silicon material is incident on the avalanche multiplication region, and the electrons in the valence band of the device will absorb the photon energy to generate energy level transition, forming free electron–hole pairs. Under the excitation of the avalanche multiplication effect, the photogenerated carriers will continue to collide with the lattice, breaking the covalent bond and resulting in the originally fixed valence electrons becoming free electrons. Through the above reciprocating process, the device can realize the macroscopic current response under the incident condition of one photon.

HDEPD adopts the polysilicon ring gate with a short anode and the concentration gradient of DNW as the guard ring to prevent the edge breakdown of the depletion layer. Due to the Gaussian distribution of DNW during ion implantation, its central concentration peak is much higher than the upper and lower sides and shows the gradient decreasing. Therefore, the DNW concentration near the top oxide layer of the device is lower, meaning that the breakdown threshold of P+ edge and DNW must be higher than the planar PN junction composed of P+ and NW in order to realize the basic function of the guard ring. In addition, the polysilicon gate with a short-circuited anode is at a low level relative to DNW, and the gate can produce a vertical upward electric field effect. The local electron concentration on the DNW surface is affected by the electric field force. Specifically, the holes can gather below the gate under the influence of the electric field, while the electrons are repelled to the bottom. As a result, the electron concentration at the edge of the P+ injection region in the center of the device is further reduced, which causes the electric field effect to focus on the photosensitive planar junction between P+ and NW, and effectively improves the quantum efficiency of the device. HDEPD adopts fully wrapped DNW as the dummy guard ring, which can also suppress the dark count caused by substrate noise carriers crossing to the multiplication region, reducing the contribution of thermal noise to DCR. The device uses P+ and NW to form the photosensitive region, the depletion layer is close to the surface of the device, and the spectral response range is close to 400 nm~600 nm, meeting the excitation spectrum detection requirements of Cd (II).

## 3. Simulation Analysis of the High-Detection-Efficiency Photodiode

As a semiconductor simulation tool, TCAD can be used to carry out one-dimensional and two-dimensional simulations according to the process and structure of HDEPD devices. Firstly, the electric field distribution and impact ionization distribution of the device working in Geiger mode were simulated to verify its basic operating principle. Secondly, the current distribution of the device under light-free and light conditions was compared to verify its conduction path and photomultiplier effect. Finally, the spectral response of HDEPD was qualitatively verified by scanning light sources with different wavelengths.

When HDEPD works in Geiger mode, the cathode is at a high potential, and the anode is at a low potential. The two-dimensional impact ionization distribution and one-dimensional impact ionization tangent distribution of the device are shown in Figure 2a,b, respectively. Meanwhile, the two-dimensional electric field distribution and one-dimensional electric field tangent distribution are shown in Figure 3a,b, respectively. According to the test results, it can be found that the peaks of the impact ionization distribution and electric field distribution of the device are concentrated in the central area of the photosensitive plane PN junction. The electric field force gathered by the side junction of P+ is small, and the guard ring can effectively prevent avalanche breakdown at the edge of the depletion layer, meaning that the device can work stably in Geiger mode to detect single-photon signals. In addition, the value of electric field force remains at 10^5^ V/cm, indicating that the turn-on of the device depends on the avalanche breakdown rather than the tunnel breakdown. The current distribution of HDEPD under light-free conditions is shown in Figure 4a. The formation of the current path includes the cathode N+ injection region, NW, DNW, central NW and anode P+ injection region, and the peak value of current density is 2.81 × 10^4^ A/cm^2^. When a photon signal with a wavelength of 600 nm is incident on the photosensitive region, as shown in Figure 4b, the current path of the device remains unchanged, and the peak value of the current density increases to 5.24 × 10^4^ A/cm^2^. At this time, the total current density is the superposition of dark current and photocurrent. Whether it is the avalanche effect caused by dark carriers or the avalanche effect caused by photons, the current will reach the saturation state after doubling, as shown in Figure 5. The simulation results obtained for the one-dimensional I–V curve show that the dark current of the device in the linear region remains at 10^−11^ A under light-free conditions and surges to 10^−8^ A under light conditions. Both the two-dimensional current simulation results and the one-dimensional I–V simulation results of HDEPD verify the photomultiplier effect of the device. Finally, by coupling a photon signal with a wavelength of 100 nm~1200 nm to the photosensitive region and recording the current response under different wavelength conditions, the spectral simulation results of the device can be obtained, as shown in Figure 6. The coordinate position of the depletion layer of the device determines its spectral response range. The spectral peak of the avalanche multiplication region composed of P+ and NW is located at 400 nm~600 nm; that is, the simulation results conform to the design principle of the device.

## 4. Experiments and Discussion

The HDEPD device is manufactured based on a 0.18 μm standard CMOS process, and the device can realize system integration without any mask correction. By establishing a quenching circuit junction spectroscopic detection instrument, the I–V characteristics, photon detection probability and dark count rate of the device can be tested. The diameter of the active region of HDEPD is 20 μm, and the layout and microscope images of the device are shown in Figure 7a,b, respectively.

The measured I–V curve of HDEPD at room temperature (20 °C) is shown in Figure 8. The test conditions of the device are divided into two groups, the first group is light-free condition, and the second group is light condition (the wavelength of light source is 600 nm, and the optical power is 10 W/μm^2^). The avalanche breakdown voltage of HDEPD is 12.15 V, and the current density in the linear region is maintained at 10^−11^ A. After the avalanche effect is generated by dark carriers, the current density is doubled to the order of mA. When the light source is incident on the photosensitive region of the device, several photo-generated electron–hole pairs are excited by the incident photons, resulting in an increase in the magnitude of the current in the linear region of the device to 10^−8^ A. The measured results are completely consistent with the simulation results. In addition, the avalanche effect caused by photon signal also make the current density reach saturation state. The avalanche breakdown voltage of the device is not affected by the light conditions and is always maintained at 12.15 V. The I–V test results of HDEPD show that the gain of photocurrent is very stable in the linear region. The reason is that the polysilicon gate shorting the anode can further reduce the surface electron concentration of the dummy DNW guard ring so that the electric field force is concentrated in the central region of the photosensitive plane PN junction. When the electric field force tends to be stable, the orderly entry of photon signals can effectively realize the photoelectric conversion function of the device.

The photon detection probability of HDEPD can be obtained by a spectrometer combined with a quenching circuit. The test instrument is a xenon light source, and the wavelength range covers the visible light band. The resolution of the point light source is 1 nm, and the incident area of the light source is 1 mm^2^, which fully meets the size requirements of the photosensitive surface of the device. When the ambient temperature is 20 °C, the device is placed in the test bench and operated in Geiger mode with external electrical connections. In order to minimize the influence of irrational factors on the probability of photon detection, the current value of the device under excess bias voltage is first tested in a dark environment, and then its current value under light conditions is tested. Since the total current of the device includes dark current and photocurrent, DCR cannot be completely suppressed. Therefore, the final processed current value is the difference between the total device current and the dark current. At this time, when the optical power is constant, the PDP of the device at different wavelength conditions can be calculated, as shown in Figure 9. When the reverse excess bias voltage of HDEPD is 1 V, the device can achieve a 31% PDP in the wavelength range from 440 nm to 600 nm, and a peak of 41.7% at 500 nm. As the excess bias voltage increases to 3 V, the PDP of the device increases to 36.8% in the wavelength range of 440 nm~600 nm, and a peak of 52.8% at 500 nm.

HDEPD adopts a passive quenching circuit combined with an oscilloscope, which can be used for DCR tests. The reverse excess bias voltage range of the device is 0~3 V, and the incremental step is 0.2 V. The dark count of the device at room temperature (20 °C) can be quantified by capturing the frequency peak of the avalanche pulse. In order to suppress the influence of noise signals on the experimental results, 10 groups of DCR under different bias conditions were tested, and the results were taken as the average, as shown in Figure 10. When the reverse excess bias voltage was 1 V, the DCR of the device was 1.8 Hz/μm^2^. When the reverse excess bias voltage was 3 V, the DCR of the device increased to 31.9 Hz/μm^2^. The test results show that the DCR had an exponential upward trend with the gradual increase in the voltage. HDEPD had a polysilicon ring gate, which could effectively isolate the STI from the multiplication region to suppress the energy level capture effect caused by material defects. In addition, wrapping the active region of the device with DNW could also isolate the contribution of substrate noise carriers to dark counts. Finally, on the premise of a lower breakdown voltage, HDEPD achieved a 52.8% PDP at the wavelength of 500 nm, and the DCR was 31.9 Hz/μm^2^. The above test data fully meet the detection requirements of the target application and can effectively improve the detection accuracy of the fluorescence spectrum analyzer and realize the high-precision measurement of Cd (II). Table 1 shows a data comparison between the detection efficiency of HDEPD and the reference device, further validating its high detection efficiency characteristics.

## 5. Conclusions

In this work, in order to achieve the high-precision measurement of Cd (II), a photodiode device with a high detection efficiency was realized based on a 0.18 μm standard CMOS process. TCAD simulation was used to verify the basic operating principle of the device. The key electrical parameters of HDEPD were obtained by building a quenching circuit and spectral test platform. The results show that the avalanche breakdown voltage of the device was 12.15 V. When the excess bias voltage was 3 V, the HDEPD could achieve a 36.8% PDP in the wavelength range of 440 nm~600 nm and a spectral peak of 52.8% at 500 nm. In addition, since the polysilicon ring gate isolated the STI from the depletion layer, the energy level trapping effect was suppressed and the DCR of the device was effectively reduced to 31.9 Hz/μm^2^. In summary, the HDEPD devices compatible with standard microelectronic processes fully meet the spectral detection requirements of Cd (II) and can provide an efficient structural design scheme for the front-end optoelectronic devices of high-precision measurement fluorescence spectrometers.

## Figures and Tables

**Figure 1 sensors-22-05630-f001:**
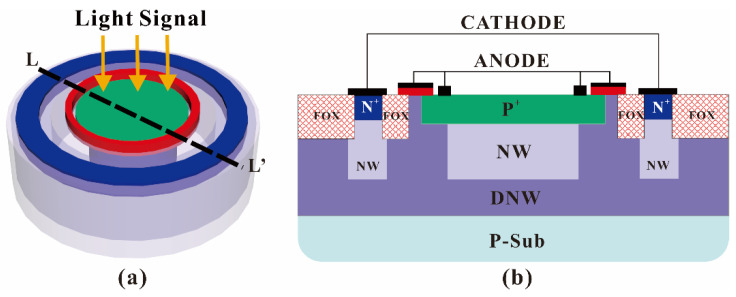
Three-dimensional schematic structure of HDEPD device (**a**) and cross-section of HDEPD device (**b**).

**Figure 2 sensors-22-05630-f002:**
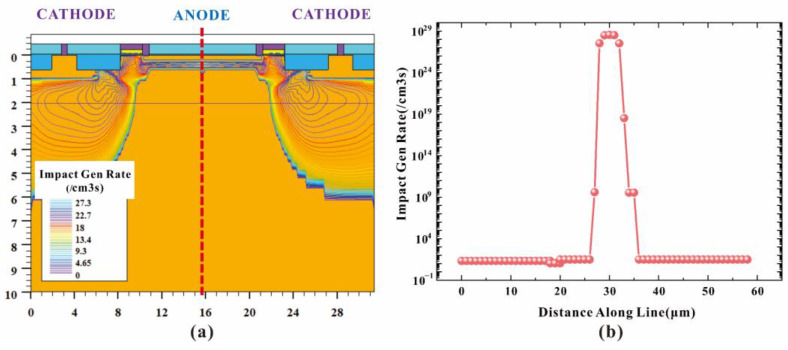
Two-dimensional impact ionization distribution of HDEPD device (**a**) and one-dimensional tangent distribution of impact ionization for HDEPD device (**b**).

**Figure 3 sensors-22-05630-f003:**
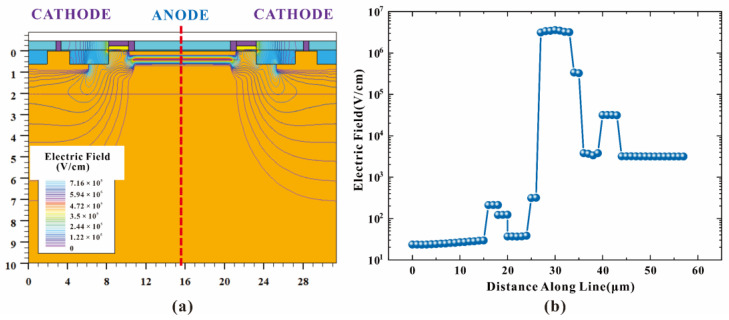
Two-dimensional electric field distribution of HDEPD device (**a**) and one-dimensional tangent distribution of electric field in HDEPD device (**b**).

**Figure 4 sensors-22-05630-f004:**
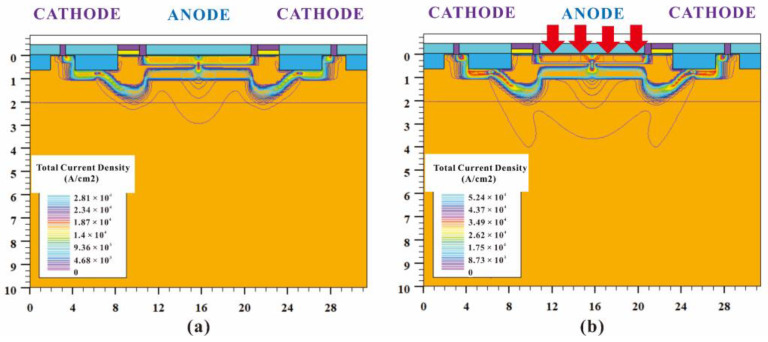
Current distribution of HDEPD device under light-free conditions (**a**) and current distribution of HDEPD device under light conditions (**b**).

**Figure 5 sensors-22-05630-f005:**
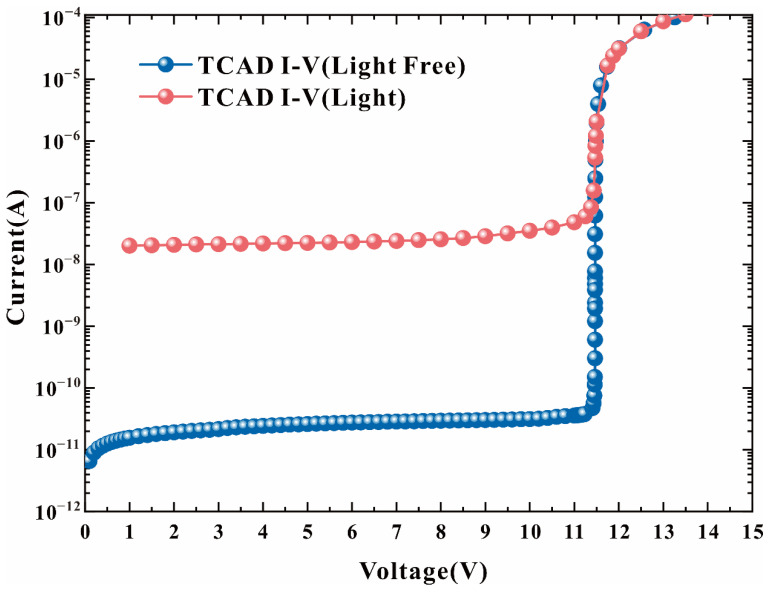
I–V curves of HDEPD device under light-free and light conditions.

**Figure 6 sensors-22-05630-f006:**
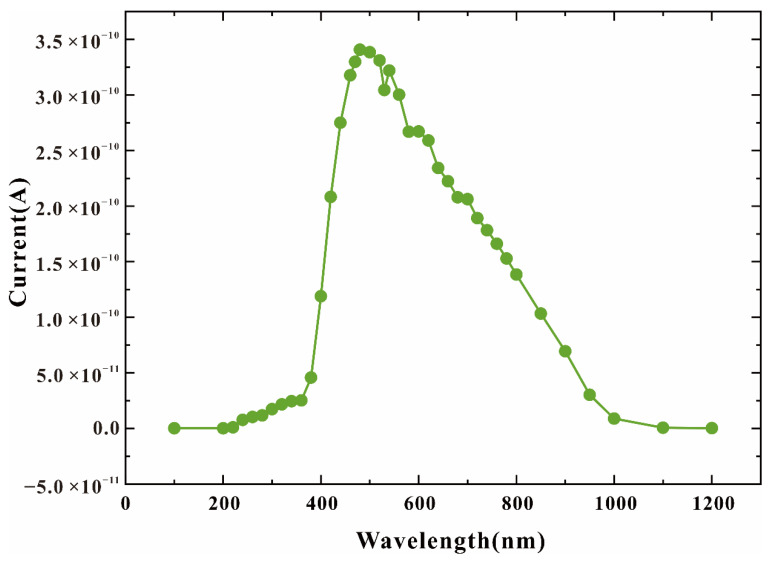
Spectral simulation curve of HDEPD device.

**Figure 7 sensors-22-05630-f007:**
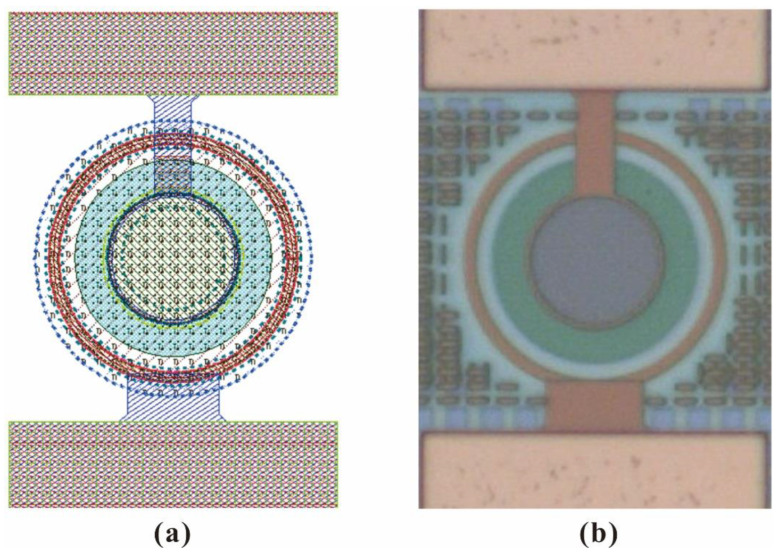
The layout of HDEPD device (**a**) and the microscope image of HDEPD device (**b**).

**Figure 8 sensors-22-05630-f008:**
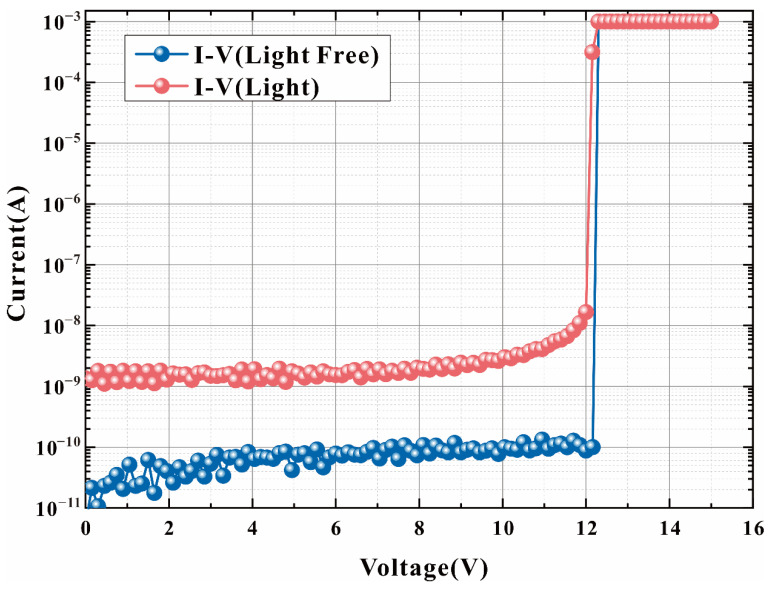
Measured I–V curve of HDEPD device (temperature: 20 °C).

**Figure 9 sensors-22-05630-f009:**
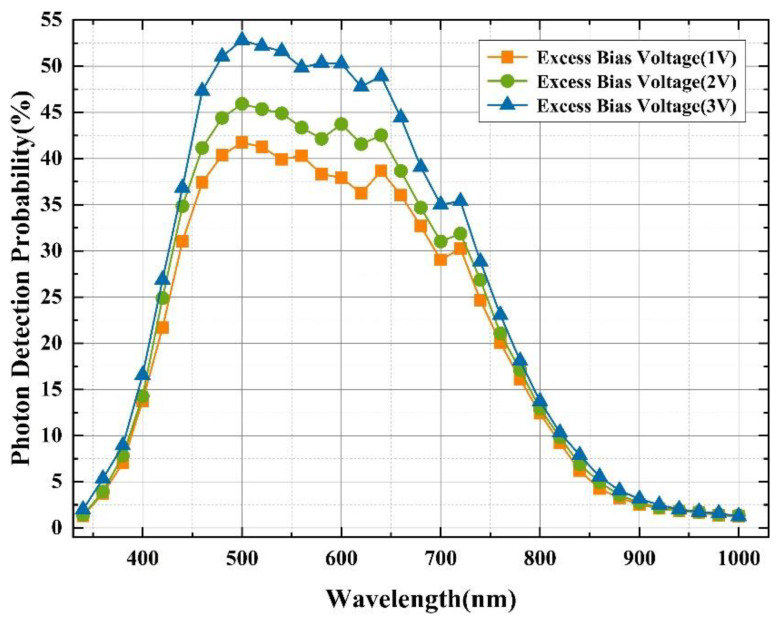
Measured PDP curve of HDEPD device (temperature: 20 °C).

**Figure 10 sensors-22-05630-f010:**
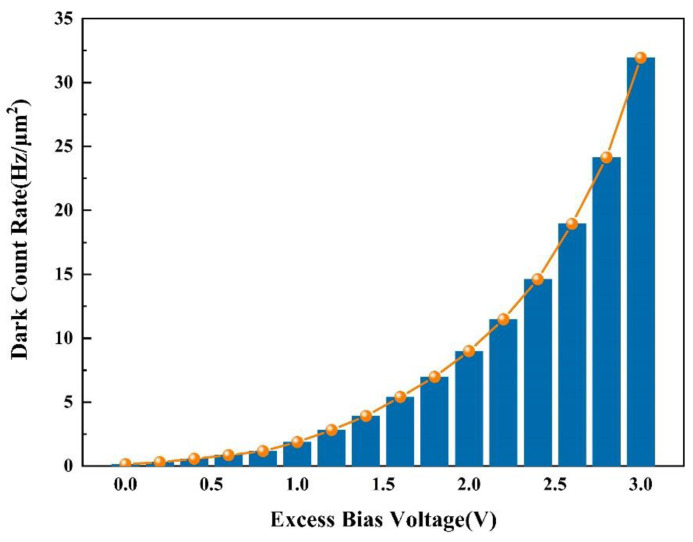
Measured DCR of HDEPD device (temperature: 20 °C).

**Table 1 sensors-22-05630-t001:** Comparison of our work to HDEPD device and reference devices.

Reference	Dark Count Rate	Photon Detection Probability
Johan, B. [35]	1.5 Hz/μm^2^ (11 V)	40%~500 nm (11 V)
Kang et al. [36]	0.31 Hz/μm^2^ (5 V)	17.5%~565 nm (5 V)
Gersbach et al. [37]	30 Hz/μm^2^ (4 V)	36%~600 nm (4 V)
Webster et al. [38]	0.5 Hz/μm^2^ (0.6 V)	28%~500 nm (1.4 V)
This Work	1.8 Hz/μm^2^ (1 V)	41.7%~480 nm (1 V)

## Data Availability

Not applicable.

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
