# Peer review of "A High-Detection-Efficiency Optoelectronic Device for Trace Cadmium Detection"

_sensors, 2022, doi:10.3390/s22155630_

Round 1

Reviewer 1 Report

The authors Huang-ling Gu and Long Wang have submitted a manuscript entitled "A high detection efficiency optoelectronic device for trace Cadmium detection" to the journal Sensors of MDPI.

The introduction provides sufficient background and includes all relevant references. The research design is appropriate. The methods are adequately described. The results are clearly presented (but the sensing part is very poor and should be improved). Discussion of data and conclusions are adequately supported by the results (but the sensing part is very poor and should be improved).

English language and style are minor spell check required.

I do not detect plagiarism and I do not detect inappropriate citations.

In general, I do not see any ethical issues along the manuscript.

In terms of originality, significance of content, quality of presentation, scientific soundness, interest to the readers, I think that the manuscript can be accepted for publication after some revisions.

1) In the title "A high detection efficiency optoelectronic device for trace Cadmium detection" should be "A high detection efficiency optoelectronic device for trace cadmium detection".

2) Chapter 2 title "Structure and Operating principle of HDEPD device" should be "Structure and Operating principle of the high detection efficiency photodiode".

Also, Chapter 3 title "Simulation analysis of HDEPD device" should be "HDEPD device of the high detection efficiency photodiode".

3) There are too many tick labels in Figure 2b and font size should be increased.

4) Font size of Figure 3 b is too small.

5) The cadmum detection discussion is very poor. The authors should show some data. Then, they should talk about possibli limit of detection, selectivity and reversibility.

Author Response

Dear Reviewer:

Thank you for your letter and for the comments concerning our manuscript entitled “A high detection efficiency optoelectronic device for trace Cadmium detection” (ID: sensors-1824242). Those comments are all valuable and very helpful for revising and improving our paper, as well as the important guiding significance to our researches. We have studied comments carefully and have made correction which we hope meet with approval. Revised portion are marked up using the “Track Changes” function in the paper. The main corrections in the paper and the responds to the comments are as following:

Responses to the comments of Reviewer #1 (C, Comments; R, Response):

C: 1. In the title "A high detection efficiency optoelectronic device for trace Cadmium detection" should be "A high detection efficiency optoelectronic device for trace cadmium detection".

R: Thanks very much to the reviewer for this suggestion. We have changed the title of the paper to “A high detection efficiency optoelectronic device for trace cadmium detection”. See Line 1-3.

C: 2. Chapter 2 title "Structure and Operating principle of HDEPD device" should be "Structure and Operating principle of the high detection efficiency photodiode". Also, Chapter 3 title "Simulation analysis of HDEPD device" should be "HDEPD device of the high detection efficiency photodiode".

R: Thanks very much to the reviewer for this suggestion. We have changed the Chapter 2 title and Chapter 3 title to “Structure and Operating principle of the high detection efficiency photodiode” and “HDEPD device of the high detection efficiency photodiode”, respectively. See Line 111-112, 174.

C: 3. There are too many tick labels in Figure 2b and font size should be increased.

R: Thanks very much to the reviewer for this suggestion. We have modified Figure 2b, removing redundant ticks and increasing the font size. See Figure 2b.

C: 4. Font size of Figure 3 b is too small.

R: Thanks very much to the reviewer for this suggestion. We have modified Figure 3b, increasing the font size. See Figure 3b.

C: 5. The cadmium detection discussion is very poor. The authors should show some data. Then, they should talk about possible limit of detection, selectivity and reversibility.

R: Thanks very much to the reviewer for this suggestion. We have modified the introduction of cadmium detection methods, and the limit of detection, selectivity and reversibility of cadmium detection have also been added in the introduction. See Line 35-70.

We would like to thank you again for the deliberation and help. We think these suggestions are very meaningful!

With best regards,

Yours sincerely,

Long Wang

Reviewer 2 Report

The manuscript is well organized and the summary is well supported by the results. Overall, I suggest it can be accepted for publication in this journal current form.

Author Response

Dear Reviewer:

Thank you for your letter and for the comments concerning our manuscript entitled “A high detection efficiency optoelectronic device for trace Cadmium detection” (ID: sensors-1824242). Those comments are all valuable and very helpful for revising and improving our paper, as well as the important guiding significance to our researches. We have studied comments carefully and have made correction which we hope meet with approval. Revised portion are marked up using the “Track Changes” function in the paper. The main corrections in the paper and the responds to the comments are as following:

Responses to the comments of Reviewer (C, Comments; R, Response):

The manuscript is well organized and the summary is well supported by the results. Overall, I suggest it can be accepted for publication in this journal current form.

R: Thanks very much to the reviewers for their comments and assistance with this manuscript.

We would like to thank you again for the deliberation and help.

With best regards,

Yours sincerely,

Long Wang

Reviewer 3 Report

Current manuscript entitled “A high detection efficiency optoelectronic device for trace Cadmium detection” by “Gu et al” proposed on the high detection efficiency photodiode which can effectively improve the detection accuracy of the analyzer. The volt-ampere curve, spectral response and noise characteristics of the device are obtained by constructing a test circuit combined with a spectral detection system. The test results show that the threshold voltage of 12.15V. The work seems good and interesting. Manuscript prepared well and can be accepted after addressing the following comments.

1.      In the introduction, authors discussed about the on-site real time detection. i.e., on page lines 37-40 “so it is difficult to realize on-site real-time detection. For your information, recently many research and review articles have been published on the on-site detection of contaminants and pollutants.

https://doi.org/10.1016/j.ccr.2021.214305

https://doi.org/10.1016/j.trac.2021.116488

https://doi.org/10.1016/j.trac.2022.116653

Note: These articles are given, for your reference only to increase the readability of your introduction. The authors no need to cite these articles.

2.      In the introduction discussion when discussing about the any reported paper, no need to write as “K. H. Kim et al.”. It should be written as “Kim et al.” Follow this throughout the manuscript.

3.      In the introduction section, discus on the related to the scope of the work. Moreover, references seem to be less.

4.      What are the units of electric field?

5.      “IV curves” should be written as “I-V curves”. Because some readers will confuse.

Author Response

Dear Reviewer:

Thank you for your letter and for the reviewers’ comments concerning our manuscript entitled “A high detection efficiency optoelectronic device for trace Cadmium detection” (ID: sensors-1824242). Those comments are all valuable and very helpful for revising and improving our paper, as well as the important guiding significance to our researches. We have studied comments carefully and have made correction which we hope meet with approval. Revised portion are marked up using the “Track Changes” function in the paper. The main corrections in the paper and the responds to the comments are as following:

Responses to the comments of Reviewer (C, Comments; R, Response):

C: 1. In the introduction, authors discussed about the on-site real time detection. i.e., on page lines 37-40 “so it is difficult to realize on-site real-time detection. For your information, recently many research and review articles have been published on the on-site detection of contaminants and pollutants.

https://doi.org/10.1016/j.ccr.2021.214305

https://doi.org/10.1016/j.trac.2021.116488

https://doi.org/10.1016/j.trac.2022.116653

Note: These articles are given, for your reference only to increase the readability of your introduction. The authors no need to cite these articles.

R: Thanks very much to the reviewer for this suggestion. In view of the lack of previous understanding, We have read more literature and have a more comprehensive understanding of the current environmental pollutant detection methods, and corresponding discussions have been added in the introduction section. See Line 35-70.

C: 2. In the introduction discussion when discussing about the any reported paper, no need to write as “K. H. Kim et al.”. It should be written as “Kim et al.” Follow this throughout the manuscript.

R: Thanks very much to the reviewer for this suggestion. We have revised all citation descriptions throughout the manuscript. See Line 81, 84, 88, 95, 98.

C: 3. In the introduction section, discus on the related to the scope of the work. Moreover, references seem to be less.

R: Thanks very much to the reviewer for this suggestion. We have read more relevant literature, referenced more literature, and added relevant background discussion. See Line 35-70.

C: 4. What are the units of electric field?

R: Thanks very much to the reviewer for this question. Results obtained through TCAD device-level simulations show that the electric field is in units of V/cm. In addition, we have supplemented the units of all TCAD simulation results. See Figure 2a, 3a, and 4.

C: 5: IV curves” should be written as “I-V curves”. Because some readers will confuse.

R: Thanks very much to the reviewer for this suggestion. We have modified the “IV curves” to “I-V curves” throughout the manuscript. See Line 221, 224, 274, 278, 290, and Figure 5, 8.

We would like to thank you again for their deliberation and help. We think these suggestions are very meaningful!

With best regards,

Yours sincerely,

Long Wang

Round 2

Reviewer 1 Report

The authors Huang-ling Gu and Long Wang have submitted the revised versio of the manuscript entitled "A high detection efficiency optoelectronic device for trace Cadmium detection" to the journal Sensors of MDPI.

I think that the authors have properly replied to all the questions raised by the referee.

I did change the quality of presentation mark from average to high. Consequently, I did change the overall merit mark from average to high.

In conclusion, I think that the manuscript ca now be accepted for publication in Sensors.